# Dynamic Spatial Responsiveness in Concert Halls

**Evan Green * and Eckhard Kahle**

Kahle Acoustics, 1050 Brussels, Belgium
* Correspondence: egreen@kahle.be; Tel.: +32-2-345-10-10

**Abstract:** In musical perception, a proportion of the reflected sound energy arriving at the ear is not consciously perceived. Investigations by Wettschurek in the 1970s showed the detectability to be dependent on the overall loudness and direction of arrival of reflected sound. The relationship Wettschurek found between reflection detectability, listening level, and direction of arrival correlates well with the subjective progression of spatial response during a musical crescendo: from frontal at pianissimo, through increasing apparent source width, to a fully present room acoustic at forte. "Dynamic spatial responsiveness" was mentioned in some of the earliest psychoacoustics research and recent work indicates that it is a key factor in acoustical preference. This article describes measurements of perception thresholds made using a binaural virtual acoustics system—these show good agreement with Wettschurek's results. The perception measurements indicate that the subjective effect of reflections varies with overall listening level, even when the reflection level, delay, and direction relative to the direct sound are maintained. Reflections which are perceptually fused with the source may at louder overall listening levels become allocated to the room presence. An algorithm has been developed to visualize dynamic spatial responsiveness—i.e., which aspects of a three-dimensional (3D) Room Impulse Response would be detectable at different dynamic levels—and has been applied to measured concert hall impulse responses.

**Keywords:** auditorium acoustics; dynamics; spaciousness; spatial response; auditory perception

## 1. Introduction

### 1.1. (Dynamic) Spatial Responsiveness

Musicians and conductors consider the concert hall part of their instrument—it is a tool for musical expression, with the best concert halls enhancing the range of expression. The deliberate variation of intensity—in other words, changes in musical dynamics—is a key ingredient in most musical composition and performance. Without both strong and subtle dynamic changes, much musical and emotional intensity is lost. Playing a musical instrument with more intensity generates not only an increase in objective sound level, but also changes in the frequency spectrum of the sound.

The connection between musical dynamics and spatial impression was observed in some of the earliest research into psychoacoustics. Marshall [1] writes in 1966 that "as a property of the hall, [spatial impression] relates to loudness attributes . . . for the listener, it generates a sense of envelopment in the sound and of direct involvement with it", with similar observations being made by Keet [2] and Barron [3]. Writing in 1978, Kuhl [4] states that (translated from German) "[spatial impression] is only, if at all, present in forte passages. Musical dynamics therefore not only influence the loudness, but with a sufficient spatial responsiveness, they increase the involvement of the room in the music. For the lay-listener, who is perhaps not even aware of this effect, this spaciousness is an unconscious yet pleasant experience." In certain conditions therefore, increases in dynamics can generate an additional dimension of spatial impression for the listener—changes in the spatial impression due to changes in dynamics will be referred to here as "dynamic spatial responsiveness".

Spatial impression, apparent source width (ASW), and listener envelopment (LEV) have been studied in much detail [3,5–10], with typical approaches using either energy integrals over relatively long time intervals (for example, "early" (0–80 ms) or "late" (0 ms to infinity) time ranges) and/or large angular ranges (such as in the definition of Lateral Fraction), or considering spatial effects as static and unchanging with performance dynamics. While valuable results have been achieved, the former approach "bundles" energy from different times and directions and does not take into account the differing sensitivity of our hearing to reflections from different directions, and nor is masking factored in: when long time ranges and large angular ranges are integrated, the (potentially very important) details of what happens in those ranges remain unclear. The latter approach overlooks the importance of dynamics and its importance in the subjective acoustical quality of concert halls.

Recently, the research group at Aalto University has taken up the topic of dynamics anew, demonstrating that dynamic spatial responsiveness is related to changes in instrument frequency responses and directivity at higher dynamic levels [11–14]: due to greater binaural sensitivity to high frequencies at lateral angles, an increased spatial impression results from the additional higher harmonics that are generated when instruments are played loudly. However, little-known research published by Wettschurek [15] in 1978 demonstrates that the connection between dynamics, loudness, and spatial impression is linked to other fundamental processes in the human hearing system, in particular how reflections from different directions are masked or unmasked at different overall listening (dynamic) levels.

### 1.2. Research by Wettschurek

In Wettschurek's thesis [15], he describes an experiment to determine the perception threshold of a test reflection dependent on overall listening levels. The experiment is shown diagrammatically in Figure 1.

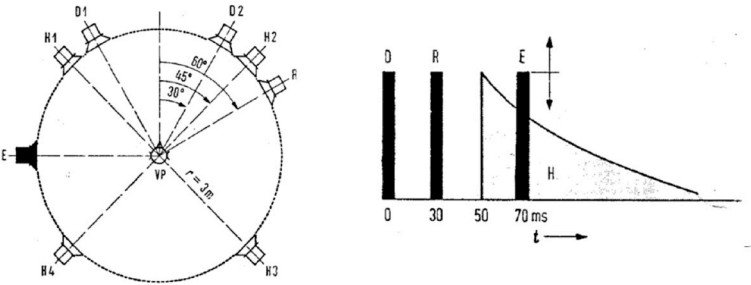

D = direct sound as phantom source via loudspeakers D1, D2
R = fixed lateral early reflection, loudspeaker R
H = 2 second reverberation (500Hz value) starting at 50ms, reproduced via 4 loudspeakers H1 – H4
E = 70ms reflection with adjustable level and location, loudspeaker E

**Figure 1.** Diagrammatic representation of an experiment to determine the perception threshold of a reflection E in the presence of direct sound D, a lateral reflection R, and reverberation H. After Wettschurek [15].

A synthetic sound field was generated in an anechoic environment consisting of the direct sound, a 2 s reverberation starting at 50 ms, and two discrete reflections: one reflection being a fixed lateral reflection at 60° azimuth and 30 ms delay, and the other a test reflection at 70 ms and of variable level and variable position. The direct sound, lateral reflection, and reverberation levels were set to achieve a direct-to-reverberant ratio of 0 dB. With anechoic speech used as source material, the overall sound level at the listening position ("listening level") was adjusted in the range 20–80 dB in 5 dB increments, subsequently the level of the test reflection was adjusted to establish the perception threshold of this reflection.

The results of this experiment, shown in Figure 2, exhibit a number of notable features:

- reflections from behind ("Hinten") have the highest perception threshold at all listening levels;
- for all reflection directions, the perception threshold decreases almost linearly with increasing listening level until approximately 40 dB;
- at listening levels higher than 40 dB, the sensitivity to reflections from the front ("Vorne") begins to plateau. Sensitivity to reflections from behind (Hinten) plateaus at levels above approximately 60 dB;
- above 40 dB however, the sensitivity to reflections from the side continues to increase approximately linearly with listening level;
- by 80 dB, the sensitivity to reflections from the side is almost 10 dB greater than for reflections from the front or rear, which at this listening level have an almost equal sensitivity of 8 dB below the direct sound level.

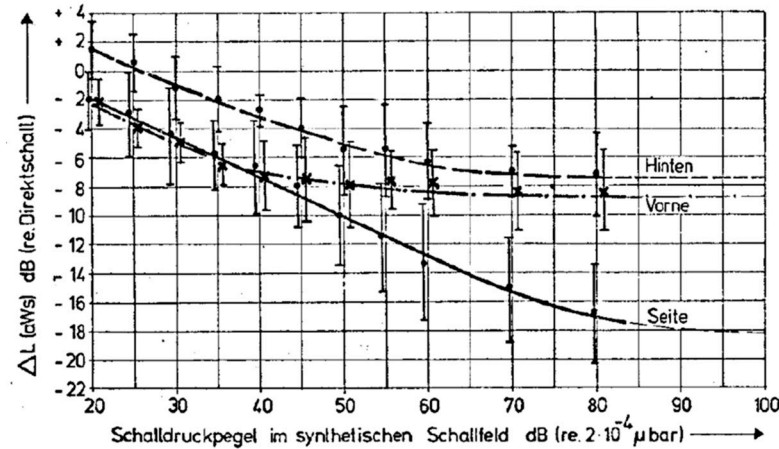

**Figure 2.** Perception threshold relative to the direct sound for a 70 ms reflection as a function of the sound pressure level at the listening position for three directions of arrival: Rear ("Hinten", azimuth 180°), front ("Vorne", azimuth 0°) and Side ("Seite", azimuth 90°) (After Wettschurek [15] Figure 1.11)).

Wettschurek's results provide a strong basis for understanding the increase of spatial impression with increasing listening level, independent of any changes in the source spectrum:

- At the lowest listening levels, most reflections would not be perceived and room presence would be extremely weak or nonexistent. The direct sound would dominate with the sources on stage being clearly localizable;
- As overall listening level increases through a crescendo, reflections from the front and side (having a lower threshold than those from the rear) would be the first to be perceived, with the expected subjective effect being an initial increase in the apparent source width (ASW), as observed and subsequently measured by Keet [2];
- As overall listening level increases further, the perception threshold for reflections from all directions of arrival continues to decrease. As a result, additional, relatively quiet reflections (e.g., from behind) become perceivable and room presence increases—this corresponds with listening experience in concert halls and the comments of Marshall [1], Kuhl [4], and others;
- Above a listening level of 60 dB, the thresholds for reflections from the front and rear plateau, while the threshold for side reflections continues to decrease. If reflection paths from the side exist, then these should become increasingly perceivable as overall listening level increases, resulting in a subjective increase in room presence.

Wettschurek refers to this journey through a musical crescendo as the room "waking up". However, this assumes that similar perception threshold relationships exist for all reflection delays and for music as well as speech. Measurements to corroborate this were not published by Wettschurek.

## 2. Reflection Thresholds with Speech and Music

### 2.1. Experiment with Speech

In order to understand how perception thresholds for reflections may vary for different delay times, directions of arrival, and for music, Wettschurek's experiment was duplicated using a binaural method. The sound field shown in Wettschurek's experiment (Figure 1) was duplicated in a virtual Ambisonics system programmed using MaxMSP and SPAT [16] software and then reproduced binaurally over open-backed headphones using the subject's own measured head-related transfer function. Head tracking was used in all experiments. Measurements of the overall listening level were made under the headphones using a calibrated sound level meter.

In a first step of the experiment, participants were asked to increase the level of the test reflection until an audible difference in the sound quality occurred (this could be any change in timbre, spaciousness, loudness, etc.). In the second step, the reflection was first set to be clearly audible and then reduced in level until the presence of the reflection could not be detected. The thresholds determined via these two steps were then averaged, but were generally within 1–2 dB of each other. Two participants carried out the following experiments, one with significant experimental listening experience, the other with lesser experience. The results shown are the average of the two participants—as has been found in previous threshold experiments, the differences found between participants were small, of the order 2–3 dB. Although the number of subjects was small, Barron [3], for example, has previously found that using small numbers of subjects (sometimes only two [3]) yields valid results.

For the first experiment, it was attempted to reproduce Wettschurek's experiment as closely as possible using anechoic speech and a 70 ms delayed test reflection. The resulting thresholds are shown in Figure 3, with Wettschurek's measurements (as per Figure 2) adjacent and to the same scale for reference.

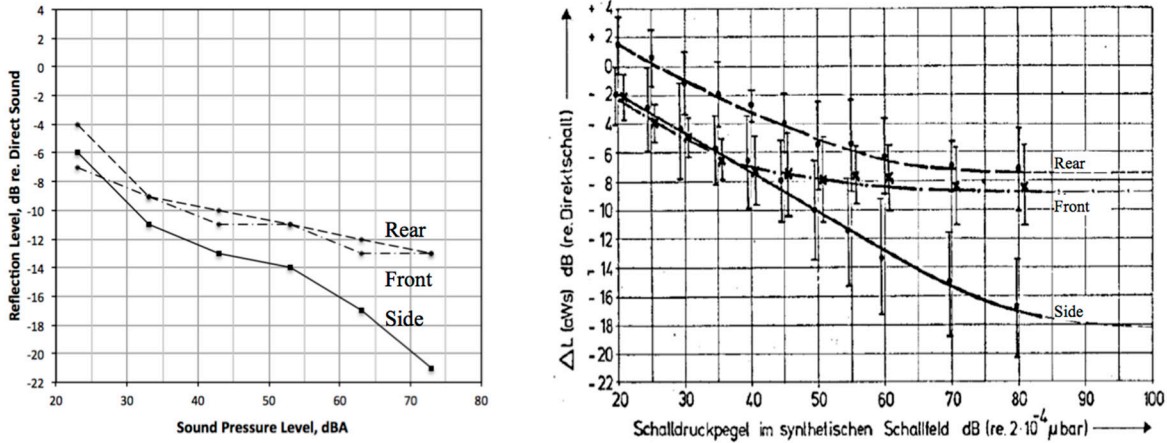

**Figure 3.** Reflection perception thresholds (reflection level re. direct sound) for speech for a 70 ms delayed reflection. Left: measured using binaural reproduction. Right: measured using a loudspeaker system (after Wettschurek [15]).

Although the perception thresholds with binaural reproduction are lower than those measured by Wettschurek, the result is encouraging, as the general relationship between the frontal, rear, and side reflections is similar. There are, however, clear differences: in the binaural case, the rear reflection threshold follows the front reflection closely—this could be attributed to known issues of median

plane confusion with binaural reproduction. Furthermore, it is known that the source material can have a significant effect on measured thresholds, and although anechoic speech was used in both measurements, the source material was different. Nevertheless, it is clear that the thresholds determined using binaural reproduction show trends similar to the (arguably more reliable) loudspeaker tests. For reference, the absolute level of the test reflection, when listened to on its own, was in all cases significantly above the threshold of hearing, therefore the measured perception threshold is an effect of masking.

### 2.2. Reflection Perception Thresholds with Music

Experiments were carried out to determine reflection perception thresholds for music using an anechoic recording of solo cello. This source material was chosen for its even sound level throughout the recording and legato musical expression to contrast with the consonant-rich, more impulsive speech source used in the previous test. The experimental procedure and participants were as described above, and tests were made for reflections delayed by 70, 50, and 40 ms. All other settings including the fixed lateral reflection, reverberation, and direct-to-reverberant ratio were maintained as previously. The results are shown in Figure 4 below, with the speech measurement for 70 ms shown in grey in the left panel.

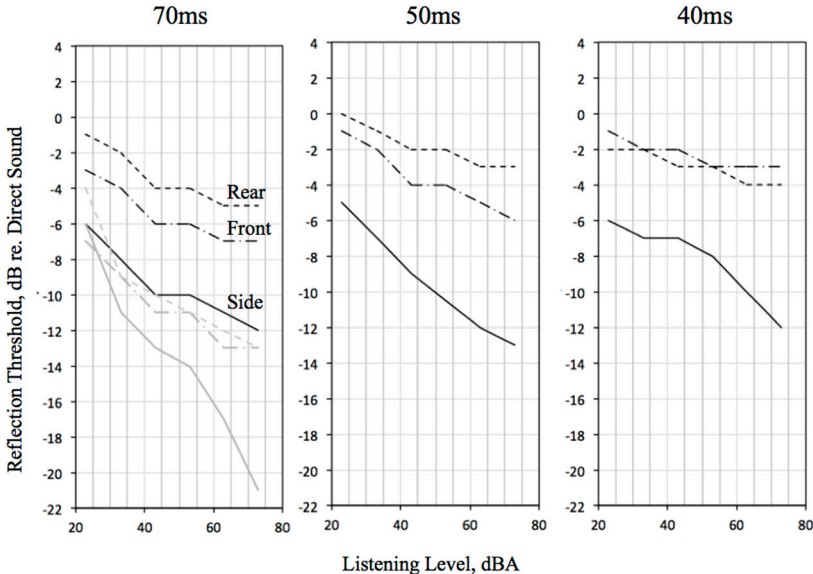

**Figure 4.** Reflection perception thresholds (reflection level re. direct sound) for music, dependent on reflection delay, reflection direction, and overall listening level. For reference, the binaurally measured reflection thresholds for speech and 70 ms delay time are shown in grey on the left panel.

Figure 4 shows that, as one might expect, the 70 ms perception thresholds are in general higher for music than for speech, reflecting the longer "cognitive integration time" for musical source material. The 70 ms reflection thresholds for side reflections are nevertheless 4–5 dB below those for frontal and 6–7 dB below those for rear directions. For the higher listening levels, these are slightly less than for speech, but the difference at low listening levels (under 40 dBA) is greater with music than with speech. At 50 ms delay time, the threshold for side reflections is almost identical to that at 70 ms, while the thresholds for frontal and rear reflections are higher. In general, the sensitivity to side reflections increases by 6–7 dB over the range of listening levels measured (approx. 25–75 dB), while the increase in sensitivity for frontal and rear reflections is 4 dB at most, with an increase of only 1–2 dB at 40 ms delay time.

### 2.3. Subjective Effects of Changing Listening Level

While carrying out the perception threshold measurements, it was clear that reflections at different delay times and directions of arrival had very different subjective effects, but also that the overall listening level had an influence on the subjective quality, even when the reflection level was close to the perception threshold. Three main changes in the quality of the sound were perceived: (1) tone coloration, (2) changes to the source spatial extent (reflection experienced as "fused" with the source), and (3) changes to the room spatial impression (reflection experienced as separate to the source).

It has long been known that one of the primary effects of reflections with short delay times is to change the tone color [3]. This was indeed the case in this experiment: when listening for whether a short-delay-time reflection was perceptible, "comb filtering" effects were often the primary quality that was identifiable. The comb filtering effect was strongest for reflections from the front and rear, while for a reflection from the side, depending on delay and listening level, tone coloration would occur simultaneously with a change in source spatial extent.

At 70 ms delay time, the subjective effect of the test reflection, once it became perceptible, was generally a change in spatial extent or spatial impression for side reflections and perceived distance for frontal and rear reflections. Most interesting, however, was the change in subjective impression of a 70 ms reflection with overall listening level. At a certain loudness level, the reflection could switch from "belonging to the source" to "belonging to the room". There is increasing evidence to support the idea that our auditory system segregates the auditory scene into various "cognitive auditory streams": a simple description of listening in musical concerts is that the auditory scene is perceived as a source (foreground) stream and a room (background) stream [17]. This experiment indicates that there are complex factors at play as to whether a sound is allocated to the source stream or room stream (or indeed to both streams, or no stream at all when below threshold): not only is delay time a factor (this experiment and others indicate that there is not a simple cut-off time at 80 ms), but also reflection direction of arrival and overall listening level. Depending on the particular combination of these factors, sounds arriving before 80 ms may not necessarily be fused into the source stream but could be segregated into the room stream. Since our subjective impressions of musical clarity, proximity, and intimacy seem to relate to aspects of stream segregation—as well as to our ability to segregate sound energy into streams at all—it is important to gain a better understanding of these relationships and auditory/cognitive processes. It should also be noted that the number of participants in the experiment described here was very small, so more research is required to understand this effect in more detail and for larger groups of listeners.

Another question that arises from this measurement is that of the effect of the "unperceived" sound energy. As mentioned above, the reflected energy which is below the perception threshold is above the absolute threshold of hearing (it is clearly audible once the direct sound and other reflected energy is switched off in the experiment). Does this "unperceived" sound energy have other subjective effects that are not picked up in this experiment, or might our hearing and cognitive system consider it to simply be noise?

One conclusion that can be drawn from the above tests with music is that, if lateral reflections are present, then as overall listening level increases one would perceive more lateral sound and the room spatial impression should change in response to the dynamics, resulting in an additional spatial dimension coupled to changes in loudness. However, if lateral reflections are weak or absent, the small differences in perception threshold with listening level for frontal and rear reflections indicate that the room spatial impression would tend to remain static: the subjective perception of the room acoustic would be less responsive to musical dynamics. Whether a dynamic or more static room presence is preferable, and whether this has a significant bearing on the overall assessment of acoustical quality is, to an extent, a matter of taste—experimental results from a much larger number of participants than were used in these experiments would be required to gain sufficient insight into this matter.

## 3. Detecting Dynamic Spatial Responsiveness

Typical objective measures related to spatial impression such as the lateral fraction (LF) and interaural cross-correlation IACC integrate lateral sound energy over a time window and so do not take into account the specific time, level, and direction of arrival of sound energy. In order see whether components of the sound field that may contribute to dynamic spatial responsiveness can be detected and visualized, and to compare measurements from different concert halls, a filter algorithm has been developed based on the thresholds measured above.

### 3.1. Dynamic Spatial Response Filter

The dynamic spatial response filter algorithm uses as its input measured first-order ambisonic B-format 3D room impulse responses (3DRIR) and processes these to reveal only the reflected sound energy that would be subjectively perceived at a given listening level. Although concert halls and music are of primary interest in this paper, since Wettschurek's experiments used a larger number of subjects and can therefore be considered to be more reliable, these reflection perception thresholds have been used in the filter. To determine the thresholds for delay times other than 70 ms and for directions of arrival other than those measured by Wettschurek, trends in our own binaural measurements and interpolation have been used.

The filtering process is shown diagrammatically in Figure 5. Firstly, the X, Y, and Z channels of the first-order B-format 3DRIR are used to calculate the direction of arrival (DOA; azimuth = θ, elevation = $\Phi$) for each sample in the impulse response—the algorithms developed at Aalto University [18] are used for this step.

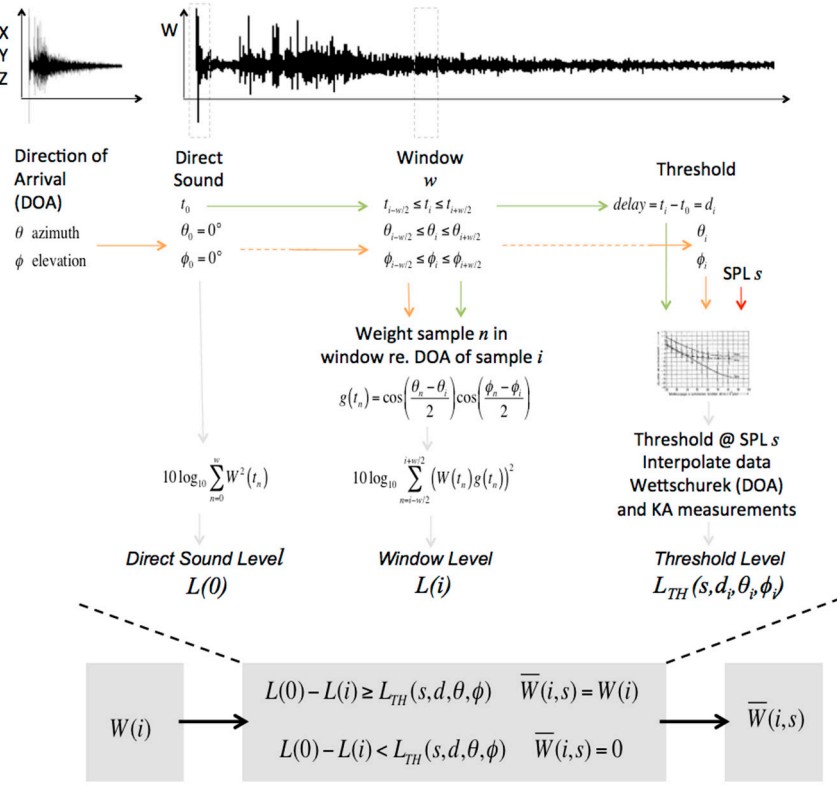

**Figure 5.** Algorithm schematic for a dynamic spatial response filter.

The direct sound level *L(0)* is determined from the omnidirectional channel *W* for a user-selected window of *w* samples. Next, for each sample *i* of the impulse response, the sound level *L(i)* is evaluated by integrating the squared impulse response over a window of the same length *w* as used for the direct

sound, centered on the sample of interest *i*. Since it is desired to collect energy in the window that can be considered to constitute a coherent reflection, each sample *n* in the window is weighted according to how similar the DOA is when compared to the central sample *i*. Therefore, before integrating the energy in the window, a weighting *g* is applied to each sample *n* in the window according to its DOA relative to the DOA of the central sample in the window. Samples with the same DOA as sample *i* are given a weighting 1, while samples arriving from the opposite DOA are weighted 0, according to a cosine rule:

$$g(t_n) = \cos\left(\frac{\theta_n - \theta_i}{2}\right)\cos\left(\frac{\varnothing_n - \varnothing_i}{2}\right) \tag{1}$$

The DOA and delay time of the sample *i* are used to establish the threshold $L_{TH}$ by interpolating the Wettschurek data and our own measurements at different delay times. If the level in the window *L(i)* relative to the direct level *L(0)* is greater than the threshold $L_{TH}$, i.e., if the sample *i* can be considered to contribute to a perceivable reflection and is not masked, then the sample *i* is passed through to the filtered output impulse response $\overline{W}$ unchanged:

$$L(0) - L(i) \geq L_{TH}(s,d,\theta,\phi) \; ; \;\; \overline{W}(i,s) = W(i) \tag{2}$$

Otherwise the sample is considered to be fully masked, i.e., the output $\overline{W}$ is set to 0 for the sample *i*:

$$L(0) - L(i) < L_{TH}(s,d,\theta,\phi) \; ; \;\; \overline{W}(i,s) = 0 \tag{3}$$

This is repeated for all listening levels of interest *s* to give a series of modified 3DRIRs, one for each listening level.

### 3.2. Results from Measurements

The dynamic spatial response filter has been applied to 3DRIRs measured in the Nouveau Siècle concert hall in Lille, France. This hall started life in the mid-20th Century as a fan-shaped conference hall (Figure 6, left) and was reconfigured in 2013, maintaining the stage and floor rake, to create a parallel-sided concert hall (Figure 6, right).

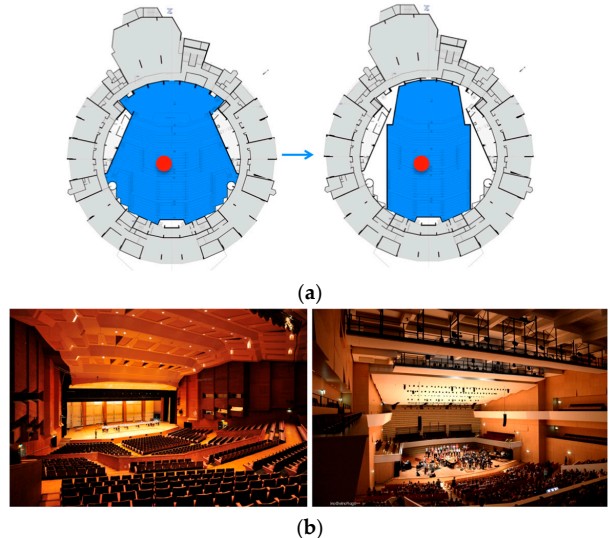

(a)

(b)

**Figure 6.** (**a**) Comparison plans of Nouveau Siècle, Lille. Left: fan-shaped conference hall before renovation. Right: parallel-sided concert hall after renovation. Measurement position for results in Figures 7 and 8 shown with red circle. (**b**) Left: Lille Nouveau Siècle before renovation. Right: after renovation with new side walls, balconies, stage enclosure, and canopy.

The fan-shaped conference hall exhibited subjective deficiencies for music uses, including a lack of reverberation, envelopment, and room presence, with the latter two attributed to an insufficient number and strength of lateral reflections. In the newly configured hall, the side walls were made parallel, while the addition of side balconies, soffits, and a modification of the ceiling design were intended to provide additional lateral reflections. Listening tests in the reconfigured hall indicate that envelopment and room presence have both been increased and that the acoustic now also exhibits greater dynamic spatial responsiveness. The question is, can these qualities be seen in the measured 3DRIRs?

Figure 7 shows measured 3DRIRs for a seat 17 m from the stage in the main seating area. The left-hand measurement is before the renovation, and the right-hand plot after. The 3DRIRs are viewed from the top, with the direct sound aligned to the "Front" direction. The arrival time is shown by color, with red shades indicating energy arriving up to 500 ms, greens in the range up to 100 ms, and blue up to 50 ms. While it is clear that after the renovation there is more energy arriving from lateral directions, indicated by the higher sound level in the "Left" and "Right" directions, can filtering help to visualize the subjectively perceived increased dynamic spatial responsiveness after renovation?

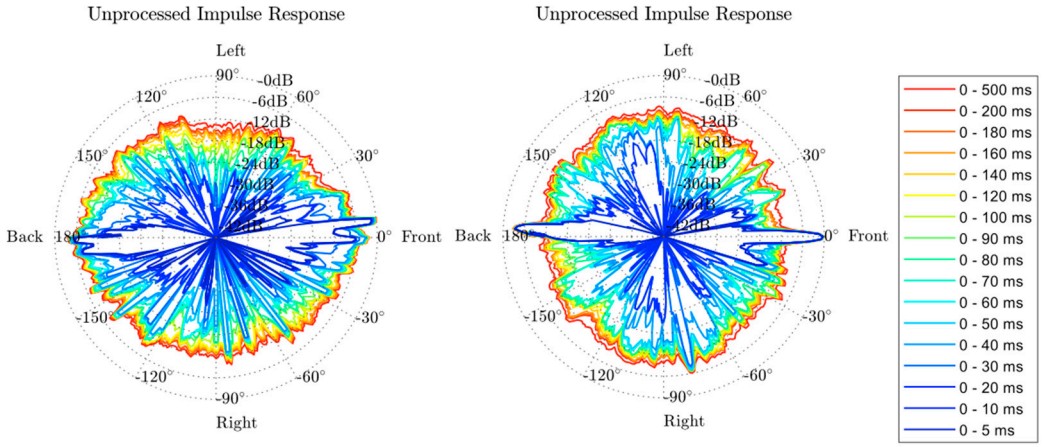

**Figure 7.** Left: unprocessed impulse response for Nouveau Siècle before reconstruction and reconfiguration. Right: unprocessed impulse response for Nouveau Siècle after reconfiguration.

Figure 8 below shows the results of applying the dynamic spatial response filter to the Nouveau Siècle measurements (Figure 7) for listening levels of 50, 60, and 70 dB and a 1 ms window length. As with the unprocessed plots in Figure 7, results from before the renovation project are shown on the left, results after renovation are on the right.

At 50 dB listening level, before renovation, perceivable sound energy is detected from the rear (likely seat reflections in the empty hall) and from the reflective stage enclosure, along with a few other "spikes" in the range 0–40 ms. Post renovation, although less sound energy would seem to be perceivable at this 50 dB listening level, a bundle of lateral sound energy from the right-hand side is clearly evident along with some later sound after 70 ms: the beginnings of noticeable source broadening and running reverberation could be expected.

At 60 dB listening level, prerenovation, the reflections already perceivable at 50 dB are supplemented by a bundle of lateral energy at around 80 ms. However, in the renovated room, the right-side reflections present at 50 dB are joined by reflections from the left-front, left-rear, and right-rear, all in the range 60–80 ms: subjectively, it would be expected that the sound is now becoming enveloping, along with a larger apparent source width.

At 70 dB listening level, the spatial response in both cases is filled out, with energy in the range 100–200 ms becoming apparent. However, in the renovated room, a new bundle of lateral energy around 50 ms from the left becomes apparent and overall the perceivable reflected energy is primarily lateral.

These plots indicate that, both before and after renovation, the perceivable room spatial impression should increase with listening level—however, it is clear that subjective room spatial impression and envelopment should be much stronger after renovation since the strength of perceivable lateral reflections is around 6 dB higher than before renovation. This magnitude of difference is not so evident in the full unprocessed impulse responses shown in Figure 7. Furthermore, before renovation the strongest reflections were clustered around the source or arrived from behind, and this attribute did not change with listening level, indicating an unresponsive room spatial impression. After renovation, the spatial zone around the source is relatively free of reflected sound energy (this can be attributed to the absorption of the new choir seating and the partially absorbing upstage wall) with the lateral directions being filled with reflected energy.

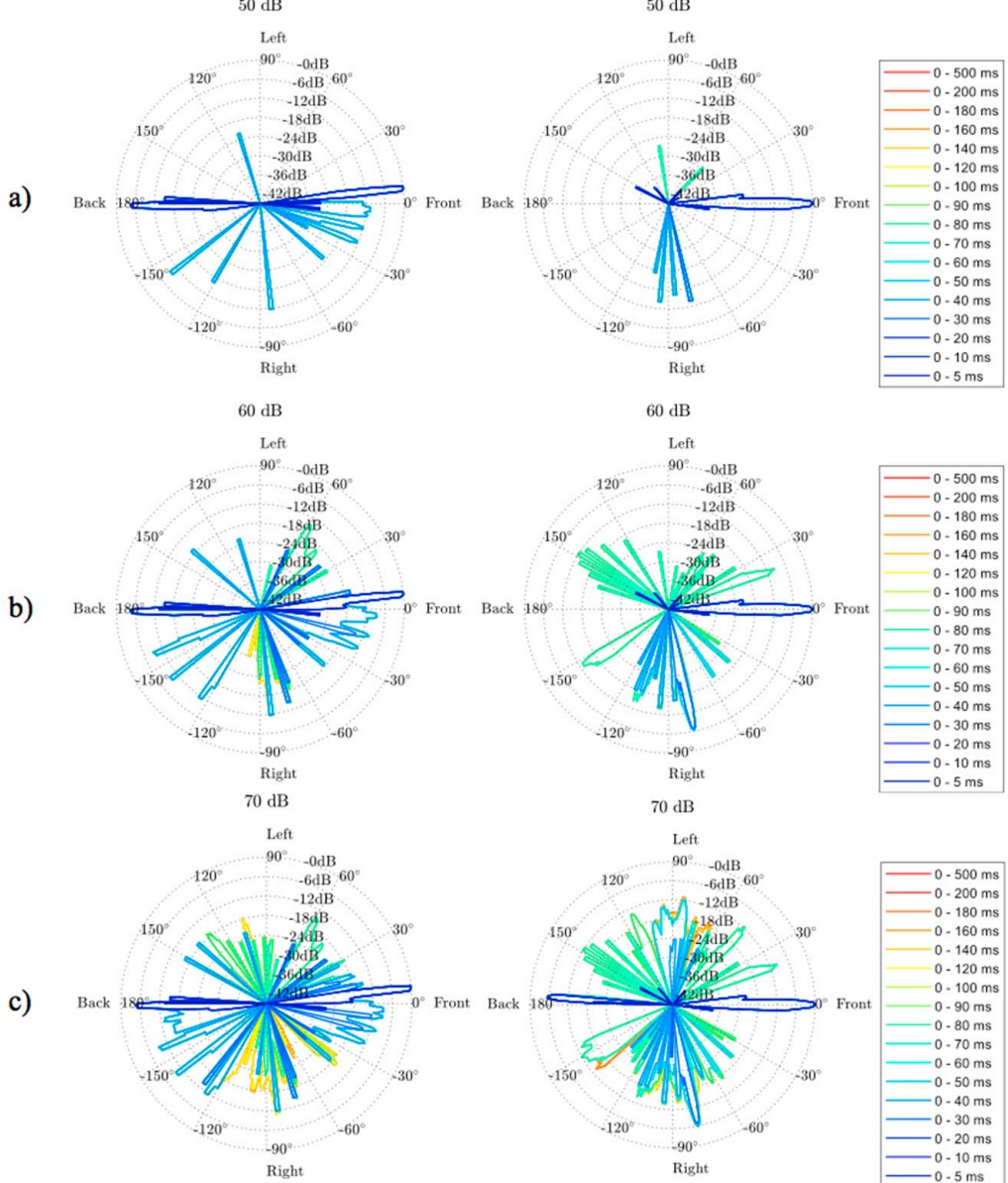

**Figure 8.** Dynamic spatial response filter applied to measurements of Nouveau Siècle, Lille for different listening levels of (**a**) 50 dB, (**b**) 60 dB, and (**c**) 70 dB. Left column: before renovation. Right column: after renovation.

Overall, through processing 3DRIRs with the dynamic spatial response filter, features of the impulse responses that indicate how the room spatial impression is perceived at different listening levels can be extracted and visualized. In the case of the Nouveau Siècle concert hall, the extracted features correspond well with the subjective effect of changing room spatial impression as loudness increases. The filter is currently relatively crude, based on a small dataset of thresholds and a very basic algorithm for attributing measured sound energy to coherent reflections. To improve the algorithm, further research is planned into the analysis of energy belonging to individual reflections and how the limited spatial acuity of the human hearing system (particularly in judging the direction of reflected energy) can be used to improve the filter.

## 4. Conclusions

Connections between musical and acoustical attributes—in this case objective loudness changes that generate subjective differences in spatial impression or room response—are believed to be an important factor in acoustical preference. Research into such connections has however been rare. Expanding on the work by Wettschurek in the 1970s, measurements of the perception thresholds of early reflections have been made using a binaural virtual acoustics system. Both speech and music were used as a source, and various reflection directions of arrival and delay times were tested. These measurements indicate that for music, perception thresholds for reflections from the front and behind vary little with overall listening level whereas for side reflections, the perception threshold decreases substantially with increasing listening level. This indicates that during a crescendo, increasing numbers of lateral reflections should become perceivable—the hall "wakes up"—while a much weaker effect is to be expected for frontal or rear reflections. These findings correspond with listening experience, with those halls lacking lateral reflections also tending to exhibit weaker dynamic spatial responsiveness.

It was also observed in the listening tests that, depending on the overall listening level, delay, and direction of arrival, reflections could be allocated to the source auditory stream (fused with the source) or experienced as part of the room stream i.e., separate from the source. This is an important factor in our perception and rating of acoustical quality, and therefore warrants detailed further research with much larger groups of listeners than were used here. The question of how the "below threshold" sound energy influences our subjective impression is also pertinent.

Based on measured reflection perception thresholds, 3D room impulse responses have been filtered in order to visualize changes in perceivable reflected energy at difference listening levels. Initial tests of the so-called dynamic spatial response filter using measurements made in the Nouveau Siècle concert hall in Lille, France indicate that such filtering can aid in the discrimination of impulse responses measured in concert halls that do exhibit dynamic spatial responsiveness. As a next step in this research, the measured thresholds and filter algorithm will be refined and applied to additional concert halls.

**Author Contributions:** Conceptualization, E.G. and E.K.; methodology, E.G. and E.K.; software, E.G.; validation, E.G. and E.K.; formal analysis, E.G. and E.K.; investigation, E.G. and E.K.; resources, E.G. and E.K.; data curation, E.G.; writing—original draft preparation, E.G.; writing—review and editing, E.K.; visualization, E.G.; supervision, E.K.; project administration, E.G. and E.K.

**Conflicts of Interest:** The authors declare no conflict of interest.

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
