# Peer review of "Dynamic Spatial Responsiveness in Concert Halls"

_acoustics, doi:10.3390/acoustics1030031_

Round 1
Reviewer 1 Report
This is a novel and interesting paper, but basic information about the psychophysical measurements (how many subjects, who the subjects were, what instructions they received, the subjects' task, how many trials were conducted) is needed.
Please insert space between numbers and their measurement units (e.g., 5 ms not 5ms)
L293, In paragraph describing Figure 8 (“Figure 8 below shows…”), need to add “and” between “left” and “results after”
Author Response
Dear Reviewer,
Many thanks for your comments.
Regarding your main point, I have added the following paragraph to explain the test procedure, number and type of participants:
"In a first step in the experiment, participants were asked to increase the level of the test reflection until an audible difference in the sound quality occurred (this could be any change in timbre, spaciousness, loudness). In the second step, the reflection was first set to be clearly audible and then reduced in level until the presence of the reflection could not be detected. The thresholds determined via these two steps were then averaged, but were generally within 1-2dB of each other.
Two participants carried out the following experiments, one with significant experimental experience, the other with minimal experience. The results shown are the average of the two participants – as has been found in previous threshold experiments, the differences found between participants was small, of the order 2-3dB. Although the number of subjects was small, Barron [3] for example has previously found that using small numbers of subjects (sometimes also only two [3]) yields valid results."
I hope that this is satisfactory.
I have made the other edits as suggested.
Reviewer 2 Report
This submission deals with "Dynamic Spatial Responsiveness" in concert halls, which is of great interest for acousticians, especially acoustical design practitioners. It provides a new method to evaluate it based on rather old work by Wettschurek back in 1970's. The work by Wettschurek is unpublished and available as a PhD thesis written in German, therefore, detailed review of this work itself in Introduction is of valuable as an information. Therefore, this reviewer suggests that this submission should be published. However, there are some points which the authors can reconsider before publication so that this article can be more understandable:
There are many subjective criteria used in the text and somewhat confusing. This reviewer advises the authors to give (even if brief explanation) a clear definition of each. For example, Spatial Responsiveness can be somewhat vague becuase its meaning can be different for some researchers. Also, "Dynamics" of musical performance is dealt with seemingly independent aspect from "Loudness" (L49), but dynamics can be the main cause of the loudness, so they are in many cases not independent. Some careful definitions and explanations can improve and avoid misleading readers.
To strengthen the statement of the novelty of this work, this reviewer advises the authors to cite more basic studies relating to spatial impression and binaural listening, i.e., spetial hearing. There have been so many papers publsihed in the effort to clarify various auditory events, e.g., spatial impression, listener envelopment and the effect of the direction of reflection on them, etc.
Relating to the comment 2, the reference cited in this submission are rather old and seems to be updated with recent studies, if there are some related publications.
Figure 5 is concise and helpful readers to understand the method intuitively, but it is better to add narrative explanation with mathematical development.
Conclusion seems to be weak. The authors "hope" ? however, they have presented results even though they are for one case. It will be better to state the target of future study to prove the generality in their sequels.
Author Response
Dear Reviewer,
Many thanks for your helpful comments.
Taking them in turn:
(1) I have adapted the introduction to more clearly introduce and define the terms that I use in the remainder of the paper, and have endeavoured to keep a consistent vocabulary throughout. In particular I have used the word "dynamics" to refer to changes in musical performance intensity, whereas "objective loudness" and "listening level" refer to the measured sound level at the listening position.
(2) I have included a paragraph referring to previous work on spatial impression, ASW and LEV, including an overview of the techniques used in previous research to provide context regarding the research presented here.
(3) More recent references are also provided.
(4) The text referring to Figure 5 has been expanded with more mathematical development.
(5) The conclusion has been adapted with a statement regarding future development. The word "hope" has been removed.
I hope that the article is now satisfactory.
Thank you again